# Formulation and Characterisation of Carbamazepine Orodispersible 3D-Printed Mini-Tablets for Paediatric Use

**DOI:** 10.3390/pharmaceutics15010250

**Published:** 2023-01-11

**Authors:** Jiayu Hu, Rawan Fitaihi, Shorooq Abukhamees, Hend E. Abdelhakim

**Affiliations:** 1Pharmaceutics Department, School of Pharmacy, University College London, 29-39 Brunswick Square, London WC1N 1AX, UK; 2Department of Pharmaceutics, College of Pharmacy, King Saud University, Riyadh 11451, Saudi Arabia; 3Department of Pharmaceutics and Pharmaceutical Technology, Faculty of Pharmaceutical Sciences, The Hashemite University, Zarqa 13115, Jordan

**Keywords:** orodispersible tablets, disintegrating tablets, mini-tablets, carbamazepine, paediatric medicine, child-appropriate formulation, 3D-printing

## Abstract

One of the main challenges to paediatric drug administration is swallowing difficulties, hindering the acceptability of the medicine and hence clinical outcomes. This study aims at developing a child-appropriate dosage form, the orodispersible mini-tablet (ODMT), using the model drug carbamazepine (CBZ). This dosage form was prepared and 3D-printed via a semi-solid extrusion technique. Design of Experiment methods were applied for optimising the formulation. The formulation with 40% (*w*/*w*) of SSG (superdisintegrant) and 5% (*w*/*w*) of PVP K30 (binder) was selected and loaded with CBZ. The drug-loaded tablets were characterised by a mean hardness of 18.5 N and a disintegrating time of 84 s, along with acceptable friability. The mean drug loading ratio of the tablets was tested as 90.56%, and the drug release rate in 0.1 M HCl reached 68.3% at 45 min. Excipients showed proper compatibility with the drug in physical form analysis. Taste assessment via an E-tongue was also conducted, where the drug did not show bitter taste signals at a low concentration in the taste assessment, and the sweetener also blocked bitterness signals in the testing. To this end, ODMTs were found to be potential candidates for child-appropriate dosage forms delivering CBZ.

## 1. Introduction

Flexible dosing is one of the main challenges in developing medicinal products suitable for the target age group(s) of interest, also known as “age-appropriate” from the Guideline on Pharmaceutical Development of Medicines for Paediatric Use established by the European Medicines Agency (EMA) [1]. In order to boost the development of child-appropriate medicines, drugs can be authorised in the EU with a new dosage form or route of administration designed for paediatric patients and apply for a paediatric-use marketing authorisation (PUMA) with market protection extension [2]. Additionally, the EMA has established the Paediatric Investigation Plan (PIP) to ensure that adequate clinical data collected to show potential acceptability in children are required for paediatric medicine authorisation [3]. While such regulatory frameworks have encouraged studies in paediatric formulation design, including designing medicinal forms and manufacturing processes for dosing personalisation, detailed guidance for developing novel dosage forms still needs to be established and standardised [4].

A new dosage form, mini-tablets, has provoked a specific interest in paediatric formulation design. Some literature characterised mini-tablets as tablets with a diameter under 5 mm [5,6]. Other literature also defined mini-tablets more precisely, with their height and diameter less than 3 mm [7]. This dosage form is introduced into the latest EMA guidelines for being child-appropriate [1]. It is suggested that by dividing the initial dose into tablets with decreased size, this formulation approach can realise a flexible dosing rationale with improved acceptability of children patients. Instead of replacing inaccurate tablet-splitting, the dose adjustment can be achieved by modifying the counting of tablets in a single dosing unit [8].

In addition to the use of mini-tablets, developing and optimising orodispersible formulations can be another strategy with great potential in paediatric medicine acceptability. The term “orodispersible” is used in the European Pharmacopeia to define uncoated formulations that disperse in 3 min upon mixture with the saliva in the oral cavity before the medicine is swallowed [9]. In the United States Pharmacopeia, such dosage forms are named orally disintegrating formulations, and the time for disintegration may range from seconds to about one minute [10]. Orodispersible formulations can be particularly beneficial in treating patients without the capability of proper swallowing, such as children or the elderly. It is noteworthy that several critical attributes need to be characterised specifically for this type of formulation, such as disintegrating time and palatability. The latter is defined in the EMA guideline as an overall appreciation of an oral formulation, considering its smell, taste, aftertaste and mouthfeel [1]. Unpleasant mouthfeel and bitter taste from the active pharmaceutical ingredient (API) or other excipients may impact patients’, and especially, children’s acceptability of the medicine, impairing their compliance to the treatment. Taste masking strategies have been applied to shelter any unpleasant taste generated by APIs and improve the palatability of medicines.

The combination of the above two strategies, results in orodispersible mini-tablets (ODMTs); these were first reported in 2011 [8]. By adapting it to hydrochlorothiazide (a diuretic drug), this novel formulation design was deemed to enable flexible dosing for children, along with improved stability and less risk of dosing errors compared with suspension products. By far, ODMT is still a novel concept in paediatric drug delivery, with limited reports and no commercially available products. In comparison with current age-appropriate formulations, ODMTs have the potential to provide precise, accurate and flexible dosing regimens to different age groups. Additionally, it applies to multiple delivery methods, including taken directly or dispersed in liquid or semi-solid for oral or nasogastric administration. Following the first report in 2011, a risperidone ODMT formulation was developed in 2015 as a needle-free alternative that is more suitable for children [11]. However, no specific discussion of taste assessment or masking was discussed, which should be regarded as a critical quality attribute considering orodispersible medicines usually expose drug particulates to taste buds. Another study reported a lorazepam ODMT, explicitly focusing on establishing a fabrication method via direct compression [12]. The formulation reported included flavouring agents to shield the bitter taste of lorazepam, yet no assessment of the masking effect was performed. In a more recent study, an electronic tongue (E-tongue) was used for taste masking assessment in an ODMT formulation [13]. An E-tongue is a device with multiple sensors which can detect different signals reflecting different taste qualities. Lipid sensors are used in E-tongue systems to transduce chemical reactions between the sensor and taste substances into electrical signals [14]. Potentiometric measurement principles are applied, and sensor responses are recorded as potential electrode values. This technology can be used in paediatric medicine development to reduce children’s participation in trials while providing reliable evidence of taste masking. The technology can reflect interactions between taste substances and is thus capable of evaluating the natural bitterness masking effect of flavouring agents or sweeteners [15].

This study aims at developing a novel ODMT formulation suitable for paediatric use. This formulation will be tested with the model drug, carbamazepine (CBZ). CBZ is an anti-epileptic drug approved with a significant clinical application in treating generalised tonic-clonic and partial seizures in epilepsy and neuropathic pain [16]. With its low solubility and its narrow therapeutic window, CBZ is often observed with variable plasma concentration in paediatric patients [17,18,19]. Currently, CBZ is authorised with a target population among all age groups, although solid dosage forms such as tablets are not recommended for children below five. This study will investigate the feasibility of adapting ODMT formulation to CBZ, along with fabricating, characterising and assessing the quality attributes of produced CBZ ODMTs. This study will specifically look at producing ODMTs with semi-solid extrusion (SSE). SSE is one of the 3D printing techniques applicable for tablet production by accumulating pastes or gels, which will form into solid dosage forms after the semi-solid preparation hardens [20]. The process of SSE for printing tablets is illustrated in Figure 1.

SSE has been applied in developing orodispersible formulation, with its simple process of preparation and fabrication [21]. The variability in printed product also makes SSE suitable for dosage personalisation, and thus it is used in this study to produce ODMTs [22]. 

This study also aimed at developing a new formulation for CBZ using the quality by Design (QbD) approach. QbD is a statistical approach to investigate the impact of different parameters, including material, formulation and process parameters on critical quality attributes (CQAs) [23]. The QbD paradigm can associate the quality of the product with its (clinical) performance and its capability of meeting patients’ needs, making it especially suitable for developing child-appropriate dosage forms. Meanwhile, physical charcterisation and other quality attributes, including drug loading, dissolution profiles and taste masking effects, of the optimized CBZ ODMTs will also be assessed in this study.

## 2. Materials and Methods

### 2.1. Materials

CBZ was purchased from Thermo Fisher Scientific (Waltham, MA, USA). Other excipients used in the formulation include lactose monohydrate from Foremost Farms (Middleton, WI, USA); Ac-Di-Sol^®^ (Grade: SD-711) from FMC Biopolymer (Philadelphia, PA, USA) and also kindly gifted from Dupont (Billingstad, Norway); Kollidon 30 from BASF (Ludwigshafen, Germany); Sucralose and D-Mannitol from Sigma-Aldrich (St. Louis, MO, USA); Glycolys^®^ (Sodium starch glycolate) kindly gifted from Roquette (Lestrem, France). Reagents used in E-tongue testing include tartaric acid, potassium chloride (KCl), potassium hydroxide (KOH), monosodium glutamate and tannic acid were purchased from Sigma-Aldrich (Dorset, UK), and silver chloride (AgCl) was obtained from Insent (Atsugi-shi, Japan). Hydrochloric acid (HCl) used in all experiments was obtained from Fisher Chemicals (Loughborough, UK). Acetone was obtained from Sigma-Aldrich (Dorset, UK).

### 2.2. Methods

#### 2.2.1. Preparation of the ODMT Formulation Based on Design of Experiment (DoE)

A paste formulation containing the same components with different concentrations was prepared for each SSE batch. Excipients used to fabricate the ODMT include lactose monohydrate as the diluent, Ac-Di-Sol as the superdisintegrant, Kollidon 30 as the binder and sucralose as the sweetener. All excipients were weighed and mixed using a mortar and a pestle. Deionised water was gradually added to the powder mixture and blended into a semi-solid paste for printing.

To optimise the formulation, the response surface design was applied as a DoE approach to investigate the impact of two factors, the concentration of superdisintegrants and binders in the formula, on the quality and performance of ODMTs (Table 1).

A central composite design (CCD) was conducted using the statistical software JMP^®^ version 15.0.0 (JMP Statistical Discovery LLC, Buckinghamshire, UK). CCD is based on a two-level, two-factor design. The trial for both factors at the middle level is repeated to fit into a second-order quadratic model for predicting responses [24]. Detailed formulation design based on CCD is shown in Table 2.

For formulation optimisation, all batches were designed as blank formulations (without drug), and the composition of each formula is shown in Table 3.

For optimised, drug-loaded formulations, 100 mg of CBZ was added on top of the 2.0 g formulation, with the amount of other ingredients remaining the same.

The DoE was completed by examining two responses, disintegrating time and hardness of printed ODMTs. The analysis of the design used a least squares fit model in JMP to predict the most suitable combination of two input factors to reach the best responses. Data were presented as prediction profilers, which can model the impact of crossed input variables, which may differ from separate observations made for each input factor [25].

#### 2.2.2. Fabrication of the ODMT Formulations via SSE

Tinkercad© (Autodesk, Inc., San Francisco, CA, USA) was used to design the model for mini-tablets 3D-printing, whereas the shape was a round tablet shape with 3 mm diameter and height. The designs were saved as stereolithography (.stl) files and exported to an extrusion-based bioprinter Bio X™ (Cellink, Gothenburg, Sweden). The paste formulations were filled into 3 mL syringes; then, a 0.41 mm plastic nozzle and compressed air were attached to the syringe. Automatic printing bed levelling and manual calibration were performed before printing to ensure batch-to-batch printing uniformity. Critical parameters are shown in Table 4.

The minimum pressure required for extrusion was recorded for each paste formulation. Unless otherwise specified, twenty-five tablets were produced for each batch and were left to dry after printing overnight at room temperature.

#### 2.2.3. Characterisation of ODMTs

A series of physical characterisations were conducted on blank ODMTs in DoE and the optimised, drug-loaded ODMTs. Notably, only appearance, measurements, weight variation, hardness and disintegrating tests were performed in DoE batches of ODMTs, while all forms of characterisation below were performed on optimised, drug-loaded ODMTs.

##### Appearance and Dimension Measurements

A total of 25 randomly selected 3D-printed mini-tablets were examined for appearance and size using a Dino-Lite Digital Microscope along with the DinoCapture 2.0 version 16.58 (Dino-Lite Europe, Almere, The Netherlands). The diameters and heights of examined tablets were measured using the scale in the capture system. Two images of a randomly selected tablet from each batch were captured using the capture system, one with the view from the top and one from the side.

The shape fidelity of each DoE batch was also used to examine the printability of the formulation [21]. In this study, the fidelity factor of printed ODMTs was calculated as the ratio of printed volume and the model volume designed in Section 2.2.2. A value closer to one represents a higher shape fidelity and a better printability of the wet mass.

##### Weight Variation

Twenty-five randomly selected 3D-printed mini-tablets were weighed using an analytical balance (VWR International, Leicestershire, UK). The mean and the standard deviation (SD) were calculated.

##### Hardness

Ten 3D-printed mini-tablets were randomly selected from each batch and tested for hardness, defined as the force required to break the tablet in the British Pharmacopeia (BP), using the Tablet Hardness Tester TBH 125 (ERWEKA GmbH, Langen, Germany) [26]. For the results of each batch, the mean value and the SD were calculated.

##### Disintegrating Time

The protocol for examining the disintegrating time for orodispersible mini-tablets was based on the method from the United States Pharmacopeia (USP) [27] with modification. Six mini-tablets from each DoE batch or the drug-loaded ODMTs were randomly selected and tested for using the Disintegration Tester ZT 34 (Copley Scientific, Nottingham, UK) in 800 mL of distilled water at 37 °C. Instead of the 2 mm mesh size in the USP apparatus, a 1 mm mesh size stainless steel wire cloth was placed at the bottom of the test tubes to make the mesh size suitable for mini-tablets testing. The time each tablet required to disintegrate fully into the bulk was recorded. For the results of each batch, the mean value and the SD were calculated.

##### Friability

The protocol for examining the friability of orodispersible mini-tablets was based on the method reported by Poller et al. [28]. Around 0.5 g of CBZ ODMTs (48 tablets) were weighed and placed into the friabilator drum (Copley Scientific, Nottingham, UK) with 6 g of 3 mm glass beads (Merck KGaA, Darmstadt, Germany) (total mass around 6.5 g, actual mass was recorded). The method for the friabilator was set with the rotation speed at 25 rpm for 100 turns. The mini-tablets, along with the beads, were collected and weighed. The friability was evaluated by calculating the percentage mass loss of mini-tablets upon testing.

##### Content Assay

ODMTs were firstly crushed into fine powders with a mortar and a pestle. Around 10.7 mg of the powder was weighed (actual mass recorded) and dissolved in the ethanol–water mixture (50% *v*/*v*), then filtered through a 0.22 μm filter (Merck KGaA, Darmstadt, Germany). The absorbance of the solution was measured twice at the wavelength of 287 nm using a Jenway 6305 UV–Vis spectrophotometer (Bibby Scientific, Staffordshire, UK), and the mean absorbance value was calculated. The testing was conducted in triplicates. The amount of drug loaded in the ODMTs was then calculated based on the calibration curve of CBZ in the solvent (linearity within 0.05–0.0005 μg/mL, R^2^ = 0.9994). All data collection and analysis were conducted with OriginPro 2021 (OriginLab Corporation, Northampton, MA, USA).

##### In Vitro Dissolution Test

A dissolution test was performed for 3D-printed CBZ mini-tablets in DT light 126 dissolution tester (ERWEKA GmbH, Langen, Germany). In the tester, each tested tablet was placed in a USP apparatus I, a basket immersed in 100 mL 0.1 M HCl media. Due to apparatus limitation, the testing was conducted in a static condition with no stirring involved. The temperature was kept at 37 ± 0.5 °C during testing. In this process, 2 mL media were collected from each vessel at 10, 20, 30 and 45 min and replaced with 2 mL fresh media. Taken samples were filtered through a 0.22 μm filter (Merck KGaA, Darmstadt, Germany). Afterwards, the sample’s absorbance was measured twice at the wavelength of 285 nm using UV-spectrophotometer (Bibby Scientific, Staffordshire, UK); the mean value was calculated [29]. The dissolution test was conducted in triplicates. CBZ percentage released from the ODMTs was calculated based on the calibration curve of CBZ in 0.1 M HCl (linearity within 1.25 to 20 μg/mL, R^2^ = 0.9999). All data collection and analysis were conducted with OriginPro 2021 (OriginLab Corporation, Northampton, MA, USA).

##### Fourier Transform Infrared (FTIR) Spectroscopy

A Spectrum 100 spectrometer (Perkin Elmer, Waltham, MA, USA) was used for obtaining FTIR spectra for all raw materials, including each excipient and CBZ, and also for the drug-loaded ODMTs (crushed into fine powders with mortar and pestle). The spectra data were collected over the wavenumber from 650 to 4000 cm^−1^, with a total of 16 scans with a resolution of 1 cm^−1^. The data were processed using the Essential FTIR v3.10.016 software (Operant LLC, Monona, WI, USA) and were plotted using the OriginPro 2021 (OriginLab Corporation, Northampton, MA, USA).

##### X-ray Diffraction (XRD)

XRD patterns for the drug-loaded ODMTs (crushed into fine powders with a mortar and a pestle) were plotted based on the data collected via a MiniFlex 600 diffractometer (Rigaku, Austin, TX, USA) supplied with Cu-Kα radiation (λ = 1.5418 Å) at 40 kV, 15 mA, over the 2θ range from 3° to 40° and at a speed of 5°/min. The exported data were plotted using the OriginPro 2021(OriginLab Corporation, Northampton, MA, USA).

##### Differential Scanning Calorimeter (DSC)

DSC thermal analyses were conducted in the drug-loaded ODMTs (crushed into fine powders with a mortar and a pestle). Circa. 5 mg samples were mounted in T130425 Tzero hermetic aluminium pans and were sealed with pin-holed aluminium hermetic lids (TA instruments, New Castle, UK). DSC analysis was performed using a Q2000 calorimeter (TA Instruments), with a nitrogen gas (flow rate at 50 mL/min). The DSC method started from 25 °C and increased at a rate of 10 °C/min up until 250 °C. The data were collected and processed via the TA Universal Analysis software version 4.5 (TA Instruments). Exported data were plotted using the OriginPro 2021(OriginLab Corporation, Northampton, MA, USA).

#### 2.2.4. Electronic Taste Sensing System Measurement of CBZ ODMTs

##### Dose Response Curve of CBZ

The E-tongue testing was established using the TS-5000Z (Insent Inc., Atsugi, Japan) with three lipid membrane sensors and two reference electrodes (New Food Innovation Ltd., Loughborough, UK). Detailed information on lipid sensors is listed in Table 5.

The preparation protocol before E-tongue testing, including sensor preconditioning and checking, is included in Appendix A. A series of CBZ in 10 mM KCl + 10% ethanol (the liquid base) solutions were prepared, with the concentration of 0.01 mM, 0.03 mM, 0.06 mM, 0.1 mM, 0.3 mM, 0.6 mM and 1 mM. The sample concentrations were prepared equivalent to intervals on the logarithmic scale, which is often used to reflect on the bitter threshold in taste assessment [30]. The liquid base was used as a blank control. The standard solution (0.045 g tartaric acid + 2.25 g KCl in 1 L deionised water) was added to all reference slots. Five sensors were attached to the measuring arm, namely C00, AE1 and AC0, and positive and negative reference sensors. The measurement cycle started with measuring reference potential (Vr) in the reference solution for 30 s and then the sample potential (Vs) for 30 s [31]. The difference between these two values reflected the initial taste of the drug. Afterwards, sensors were washed in reference solution for 2 s × 3 s and tested for potential in the reference solution (Vr1) for another 30 s. The difference between Vr1 and Vr, named Change of membrane Potential caused by Adsorption (CPA), reflected the aftertaste of the drug. Upon each sample measurement, sensors were washed in the alcohol solution for 330 s before testing the next sample. All testing was repeated four times, where the first run was regarded as conditioning and discarded. The mean and the SD of the remaining three sets of results were calculated, and the data were plotted using OriginPro 2021 (OriginLab Corporation, Northampton, MA, USA).

##### Taste-Assessment of CBZ ODMTs

Taste extracted liquids were prepared for testing the formulation [31]. For taste evaluation, one CBZ ODMT (equivalent to a single dose unit) was weighed and dissolved in 100 mL of the liquid base, and the concentration of CBZ in the solution was calculated based on the content assay result. The mixture was then filtered through a 0.22 μm filter (Merck KGaA, Darmstadt, Germany), and the remaining measurement followed a similar procedure. 

## 3. Results and Discussion

### 3.1. Fabrication, Characterisation and Optimisation of ODMT Formulation via DoE

A primary focus of this study was to investigate whether SSE was suitable for fabricating ODMTs. According to the EMA guideline, excipients used in children require toxicological data in specific age groups, and thus a formula with fewer excipients is preferable in paediatric formulation design [1]. Compared with conventional tablets, the composition of ingredients used for paste preparation was relatively simple and did not require any considerations, such as the flowability of the raw material. However, the extrusion process could be considered time-consuming. In this study, the formula could only print 50 tablets at most (as one batch), and the printing rate was observed to be similar to the literature (13–15 min for 50 tablets). While the printing may not be suitable for large-scale production, it can be used for dose personalisation, for the drug content and the number of tablets printed for each patient can be adjusted on demand.

#### 3.1.1. Fabrication of ODMT Formulation in DoE

According to the DoE, ten batches of blank ODMT formulation were produced via SSE. During the fabrication, the minimum pressure allowing proper extrusion of the semi-solid paste differed in each batch, which was recorded in Table 6.

Theoretically, the DoE based on CCD can estimate the optimal combination of the two variables within their lowest and highest levels. However, for F4, F6, F8 and F10, the semi-solid pastes were either not extruded or suffered from extrusion discontinuity due to poor viscoelastic properties, making it insufficient to build tablets even under the maximum pressure of the printer (200 kPa) [32]. The four formulations were deemed not feasible via SSE fabrication and excluded from the DoE and further characterisation.

It is noteworthy that the amount of water added to the powder mixture could significantly impact the tablet dimension, mass and also the paste consistency profiles [32]. In this study, the amount of water and the extrusion pressure were initially considered invariants to maintain controlled conditions within different formulations. However, during the actual preparation, the viscoelastic properties varied due to the different compositions of excipients. Hence in the DoE fabrication, the pressure used for extruding pastes was changed to adapt to different paste formulations. The solid: liquid ratio and the extrusion pressure were changed in further studies. These two variables should be regarded as responses to different formulation compositions.

#### 3.1.2. Characterisation of ODMT Formulation in DoE

Six batches of ODMTs were successfully produced via SSE. The representative appearance of ODMTs in each batch is shown in Figure 2. The figures were aligned as similar size to compare the appearance, so the diameter and height values are not compatible here but are recorded in Table 7.

All minitablets from six formulations showed a cylinder shape as designed, with homogeneous white colour and obvious layers as shown in the images captured from the side. From the top view images, the concentric infill pattern could be seen in tablets in F1, F3 and F4, yet not in the other three formulations. All tablets showed irregular edges, and for F2, F3, F4 and F6, printed extrusion residues around the base layer were observed that could be due to the poor adhesion of the paste to the building plate or the paste strand was torn by the next layer [33].

Six batches were then characterised in dimension and weight variation, hardness and disintegration. Data were collected accordingly, as shown in Table 7.

The height of mini-tablets in all 6 batches was less than 3 mm, which was decreased compared to the 3 mm × 3 mm model for printing. The diameter of mini-tablets varied among the 6 batches and only F9 produced mini-tablets with a diameter of less than 3 mm. According to the fidelity factors, F5 had the most proper printability as well as consistency. Calculated as mean/SD (%), the mass variation ranged from 8% (F5) to 21% (F1). Despite possible measurement errors, the variation could also result from the physical properties of different formulation compositions or the solid:liquid ratio/extrusion pressure that was hard to control in the experiment. Since the average mass of tested tablets was below 40 mg, the limitation of mass variation does not apply. The average hardness of tested batches ranged from 49.6 N (F1) to 101.2 N (F3). F3 showed the highest hardness variation (SD = 13.20) within the batch and F7 the lowest (SD = 5.18). As for the disintegrating time, only F2 could be characterised as an orodispersible formulation considering the limitation of 180 s for orodispersible tablets standardised in the BP [26]. This formulation also showed the lowest variation (SD = 8.110), while the highest variation was found in F9 (SD = 23.85). All other formulations failed to disintegrate within 180 s, which was not consistent with the result reported by Eduardo, Ana and José, where similar excipients were used in SSE-printed tablets [20]. Apart from using possibly aged superdisintegrants, another likely explanation for such difference is that the reported study had used the standard disintegrating test protocol. In the previously published studies, tablet disintegration was tested using apparatus with the 2 mm mesh size, while the diameter of the tablets was around 5 mm. Such a method could have resulted in a falsely higher disintegrating rate. Meanwhile, in this study, the actual mesh size in the disintegrating test was less than 1 mm, and the tablets tested were around 3 mm in diameter. Since no standard protocols have yet been established for the physical characterisation of mini-tablets, the results could also be impacted by the method applied for testing. Considering the dimension of mini-tablets, the disintegrating test method used in this study may be more suitable in similar studies.

#### 3.1.3. DoE Analysis and Optimisation of ODMT Formulation

Based on the initial design and collected data, the DoE analysis was conducted using the standard least squares regression analysis model. For fitting two responses separately, the summary of the fit of the model was shown in Table 8.

In order to optimise the disintegrating time and hardness, prediction profiler graphs were plotted in Figure 3, Figure 4 and Figure 5 using JMP. The red lines indicated the optimal value of each variable and response.

The *x*-axis in the graph represents input variables, and the *y*-axis represents the predicted outcomes. The desirability indicates the likelihood of an outcome resulting from the optimal input variables. The prediction profiler illustrated that the optimum combination of superdisintegrant and binder concentration (*w*/*w*) applied in the ODMT formulation for SSE fabrication should be 40% and 5%, respectively. This prediction was made under the limitation of disintegrating time within 180 s, whereas the disintegrating time of most formulations did not meet the standard of orodispersible tablets.

In order to further improve the disintegrating profiles of F2, the original superdisintegrant, Ac-Di-Sol (40% *w*/*w*), was replaced by sodium starch glycolate (SSG) (40% *w*/*w*), or D-Mannitol (30% *w*/*w*) + SSG (10% *w*/*w*) The remaining composition of F2 was not changed. Differences in fabrication methods and characterisation results of two new formulations compared with F2 are listed in Table 9 and Table 10 (appearance comparison shown in Figure 6).

For both new formulations, the mass of printed mini-tablets increased to around 20 mg, and the weight variation was decreased to less than 10%. In addition, the dimension of tablets in F2a and F2b increased, and the dimension variation of F2a was also the lowest among the three formulations. Meanwhile, both new formulations showed a better printability than F2. In the figures, tablets from F2a showed a more consistent round shape with fewer residues on the edge, while those from F2b showed a looser structure and a less consistent shape. The average hardness of the two new formulations was lower (53.3 N in F2a and 31.9 N in F2b) than F2. However, the hardness data were also more consistent within tested samples for F2a and F2b, indicated by lower SD values (6.65 and 8.00).

Since the change of superdisintegrant was mainly aimed at improving disintegrating time of the formulation, detailed statistical analyses were conducted merely on the results of this response to confirm the significance of differences. A set of F-Tests and Student t-Tests were conducted using Excel to analyse the difference of the disintegrating data among three formulations. For all statistical tests in this study, α = 0.05. Both F2a and F2b showed a significant difference in disintegrating time from F2 (with both *p* values < 0.0001, while the difference between both new formulations was insignificant (with a *p* value of 0.274 > 0.05). Based on the results, F2a showed a low disintegrating time and sufficient hardness. Thus, it was selected as the formulation with the optimum concentration of superdisintegrant and binder for CBZ ODMTs. Additionally, the composition of F2a was simpler than F2b since the latter formulation involved an additional ingredient, D-Mannitol.

Though both excipients are widely used as superdisintegrant, the disintegration mechanisms are different between SSG and Ac-Di-Sol. The former is a hydrophilic material that can uptake water and swell in the solution, while the latter is a water insoluble polymer that forms a swellable matrix in contact with water [34]. Such distinct mechanisms may affect their function in the SSE-printed tablets, which requires further investigation.

### 3.2. Characterisation of Optimised CBZ ODMTs

#### 3.2.1. Characterisation Compared with Blank Formulation

The drug-loaded formulation (coded as F) applied the composition of F2a with an additional 100 mg of CBZ. For the drug-loaded formulation, a total of 50 mini-tablets were 3D-printed for all characterisation procedures. Differences in fabrication methods and characterisation results of F compared with F2a are listed in Table 11 and Table 12 (appearance comparison shown in Figure 7).

A set of F-Tests and t-Tests confirmed that all characterisation profiles except for the disintegrating time of drug-loaded ODMTs were statistically significantly different (*p* < 0.05) from the profiles of blank placebos. In addition, F’s weight and dimension variation were lower than F2a, indicating an improved batch consistency. The printing fidelity of the drug-loaded formulation was not as proper as the blank one. In the figures, F tablets were observed with clear layers and a looser structure without any residues. One possible reason for these differences is that the solid:liquid ratio and extrusion pressure for F and F2a were different, which could impact the viscosity of extruded paste, the binding adhesion among layers and thus the physical properties of printed tablets, as mentioned before. In addition, the API itself could bring a property change to the formulation. Such differences also suggested that when considering the limitation of a dose loaded to the SSE-produced formulation, the impact of APIs should be taken into account to ensure proper physical characterisation profiles in final tablets.

The average disintegrating time of the drug-loaded ODMTs was still within the limitation of orodispersible formulation standards. Meanwhile, the average hardness of drug-loaded ODMTs was 18.5 N, the lowest value among all tested formulations. While the existing standards in the BP are only applicable for conventional tablets without specifying the tablet fabrication method, a distinct hardness acceptance may need to be established for ODMTs [26].

#### 3.2.2. Further Characterisation of Drug-Loaded ODMTs

##### Friability

The initial total mass of tablets mixed with glass beads was 6.5061 g. After the friability test, the total mass was weighed as 6.5018 g. Hence, the friability was calculated as 0.84%, which was within the acceptable limit of the BP standard of 1% [26].

##### Content Assay

The content essay results of three samples were calculated based on the calibration curve and presented in Table 13. The percentage of drug loading was calculated as the ratio of the actual value to the theoretical value (the composition of CBZ in the powder mixture). Similar to the hardness value, acceptance standardised in the BP could not be applied to the current assay to determine content uniformity of the ODMTs [26].

The theoretical dose of CBZ ODMTs was designed as 100 mg in 2100 mg of solid mixture. The final formulation determined the average actual dose as 0.4657 mg in 10.71 mg ODMTs. According to the British National Formulary for Children (BNFC), the recommended dosing regimen for children between 1 month and 11 years old starts with an initial dose of 5 mg/kg per day [35]. The dosing can be further increased by 2.5–5 mg/kg and up to 20 mg/kg daily. Hence, the relatively low dose in the ODMTs developed in this study can be potentially adapted to a flexible dosing adjustment in paediatric patients. The drug content can be adjusted to 0.5 mg per tablet, which can be used as the initial and increasing dose unit. The limitation of this preparation, however, is the consistency within the batch that could result in inaccurate dosing, which needs to be optimised by adjusting the paste formulation to reach a more homogenous texture, as mentioned before. The drug loading of 90.56 (±0.29)% was possibly due to an improper printing process, such as clogging, or intermittent extrusion where air was present in the paste, which would result in the loss of wet mass and thus the low drug content [36].

##### Dissolution Profile

The dissolution profiles of the three samples are plotted in Figure 8.

From the graph, the drug release in the first 10 min was relatively slow, while the rate of drug release was higher after 10 min. The release rate then continued to decrease, with the maximum released percentage of drug reaching 74.5% at 60 min, which conformed to the criterion of at least 70% being released from the formulation [37]. The CBZ ODMT formulation did not show a rapid dissolution profile similar to most orodispersible formulations, which is possibly due to the method applied for dissolution testing. No stirring was included during the dissolution due to the limited equipment suitable for testing mini-tablets, which may have resulted in a release more extended than the in vivo situation.

##### FTIR

The FTIR analysis data of CBZ ODMTs were plotted and compared with all raw materials in Figure 9. In comparison with all excipients in the formulation, the FTIR spectrum of the final tablet showed a pattern most similar to the pattern of raw lactose monohydrate. All peaks were identified based on the reference chart [38]. A distinct group of peaks (highlighted in a red frame) observed in the formulation’s spectrum can be related to overlapping peaks of the C-O group stretching in PVP K30, SSG and CB, and also the C=C group stretching in CBZ. No significant peak shift or new peak was observed in the spectrum, suggesting that no chemical interaction was detected in the formulation via FTIR analysis.

##### XRD

The XRD analysis data of CBZ ODMTs were plotted and compared with all raw materials in Figure 10. The diffraction pattern of the ODMT formulation was observed with sharp, distinctive peaks which indicated its crystalline form [39]. Meanwhile, a characteristic curve representing an amorphous state was also observed in the pattern, suggesting the physical form of the final formulation is partially crystalline and partially amorphous. Most of the peaks (within 10 to 35°) from the ODMT pattern were identical to the lactose monohydrate pattern, only with less intensity. A peak shift was observed at around 37°; since the major peaks from the excipients and the pure drug were shown, such a peak shift can be considered as experimental error [40].

##### DSC

The DSC analysis data of the drug-loaded ODMTs were plotted and compared with all raw materials in Figure 11. All downward peaks in the graph represent endothermic processes of the material. The curve of dried ODMTs displayed a broader peak below 100 °C, possibly reflecting a process of water loss. However, this peak was not identical to the peak of PVP K30 or SSG. Such water loss could originate from the water added to the paste bound to the formulation. The result confirmed that the relatively low drug loading ratio mentioned before was due to water partially bound in the solid structure, increasing the tablet mass. Meanwhile, two other peaks were observed in the same curve. The peak around 150 °C can be aligned with the dehydration of the lactose [41]. The peak around 201 °C can cover the melting peaks of lactose monohydrate and CBZ. However, the actual peak shifted slightly from the peaks observed in raw materials. A possible explanation for such results was the part of the water bound to the structure, which formed intermolecular forces that affected the existing endothermic processes of excipients [42]. Overall, physical form analysis of raw ingredients and the formulation revealed good compatibility between the API and excipients.

### 3.3. Electronic Taste Sensing System Measurement of CBZ ODMTs

Taste assessment in this study focused on investigating the acceptability of the drug and the palatability of the ODMTs, along with insight towards the applicability of E-tongue for CBZ formulations.

#### 3.3.1. Dose Response Curve of CBZ

A series of CBZ solutions were tested with a number of sensors to see if the drug can be detected with any signals indicating a bitter taste. The mean responses of the three sensors were plotted in Figure 12 (initial taste) and Figure 13 (aftertaste, CPA). Within the tested concentration, the drug did not yield a response beyond 5 mV, and thus the drug was considered as undetectable for the three sensors used [43]. In other words, CBZ did not show a significant taste quality of bitterness or astringency [44]. CPA signals for the three sensors were also not detectable; therefore, it can be deduced that no bitterness or astringency would linger through as an aftertaste, suggesting that simple taste-masking strategies may be sufficient to ensure this ODMT is palatable.

It is worth noting that the concentration used for taste assessment here is relatively low, and a higher concentration could not be applied to the E-tongue due to the low solubility of CBZ (a solubility of 18 mg/L at 25 °C) [45]. The bitterness detected may therefore be problematic if the dosage is higher. This is especially evident if the dosing regimen requires multiple units or increased doses in a single unit, as bitterness may become more significant and taste masking approaches may need to be involved. Meanwhile, low ionisation of the drug could also be a limitation of the E-tongue assessment. The insignificant result of E-tongue is also not conclusive, and a human taste panel could be employed for further investigation of the palatability of the drug and its formulation.

#### 3.3.2. Taste-Assessment of CBZ ODMTs

The CBZ ODMT formulation was tested similarly to the pure drug solution. The concentration of CBZ in the formulation solution was calculated to be 0.02 mM, or approximately 0.5 mg/100 mL. The relatively low dose tested here was, again, mainly due to the low solubility of the drug. However, the dose was clinically meaningful since the tablet was able to be used as a single unit dose, with its potential to be applied to a flexible dosing adjustment for paediatric use. Only the AC0 sensor (reflecting basic bitterness) showed valid data, suggesting that the other sensors may not be applicable for the formulation. The response of AC0 was plotted and compared with the AC0 responses to the pure drug, both with the initial taste and the aftertaste (Figure 14). From the graph, the CBZ ODMT was observed with slightly more significant signals than the pure drug, possibly due to the excipients’ interaction with the sensor. Nevertheless, all signals detected by AC0 were still within ±5 mV. Thus, the final formulation was not detectable by AC0.

## 4. Conclusions

This study confirmed the feasibility of fabricating CBZ ODMTs via SSE 3D printing. The formula with 40% (*w*/*w*) of SSG and 5% (*w*/*w*) of PVP K30 was selected and loaded with CBZ. The drug-loaded ODMTs showed acceptable physical attributes, including a disintegrating time of 84 s and a hardness of 18.5 N. A drug loading of 90.56% and a relatively sustained release of the drug was also found in the characterisation. Excipients used in the formulation showed good compatibility with the drug. A taste assessment of both the pure drug and the final formulation was conducted. The drug only showed an obvious reaction with the E-tongue sensor at a higher concentration and was most likely would not be perceived as bitter in a lower dose in paediatric use. The use of sweetener also blocked bitterness signals in the testing and proved an efficient taste masking approach for developing CBZ ODMTs as a child-appropriate formulation.

Overall, ODMT is a potential dosage form for delivering CBZ in paediatric patients. The formulation designed and fabricated in this study showed advantages in quick disintegration, fractionated dose and no unpleasant taste or mouthfeel detected in in vitro taste assessment. Methods used in this study for the disintegrating test, the friability test and the dissolution test also provided insights into establishing characterisation and quality control standards.

## Figures and Tables

**Figure 1 pharmaceutics-15-00250-f001:**
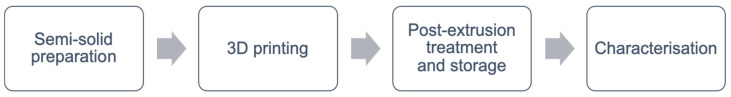
Graphic illustration of ODMT production via SSE.

**Figure 2 pharmaceutics-15-00250-f002:**
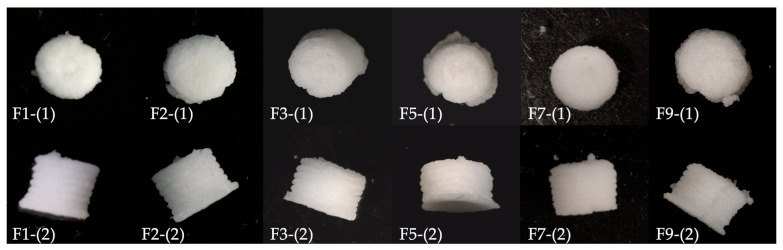
Appearance of ODMT formulation (post drying for 24 h) in DoE: F1-(1), F2-(1), F3-(1), F5-(1), F7-(1), F9-(1): top view; F1-(2), F2-(2), F3-(2), F5-(2), F7-(2), F9-(2): side view.

**Figure 3 pharmaceutics-15-00250-f003:**
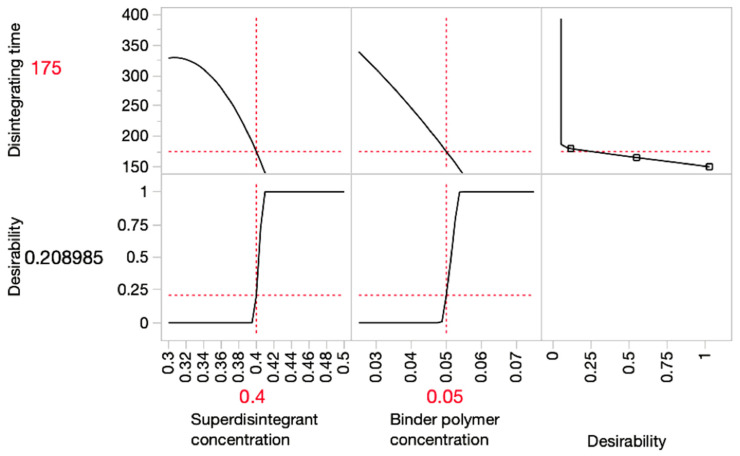
Prediction profiler graph for predicting the optimal formulation for minimising disintegrating time.

**Figure 4 pharmaceutics-15-00250-f004:**
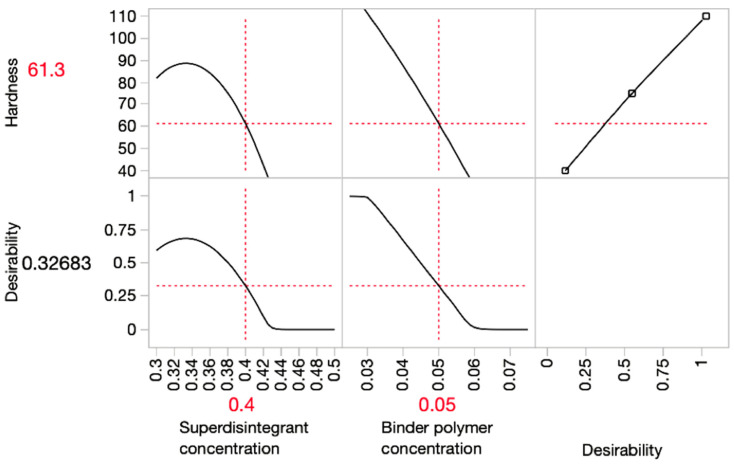
Prediction profiler graph for predicting the optimal formulation for achieving proper hardness.

**Figure 5 pharmaceutics-15-00250-f005:**
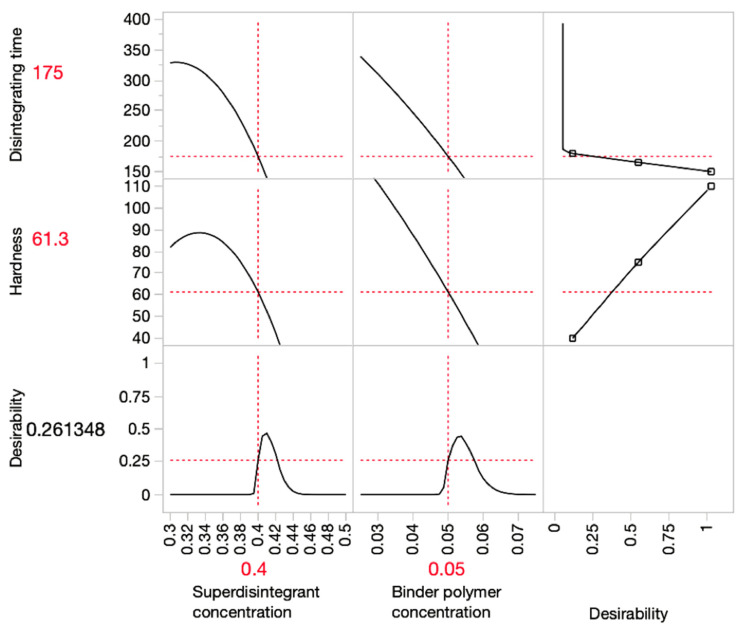
Prediction profiler graph for predicting the optimal formulation considering both disintegrating time and hardness.

**Figure 6 pharmaceutics-15-00250-f006:**
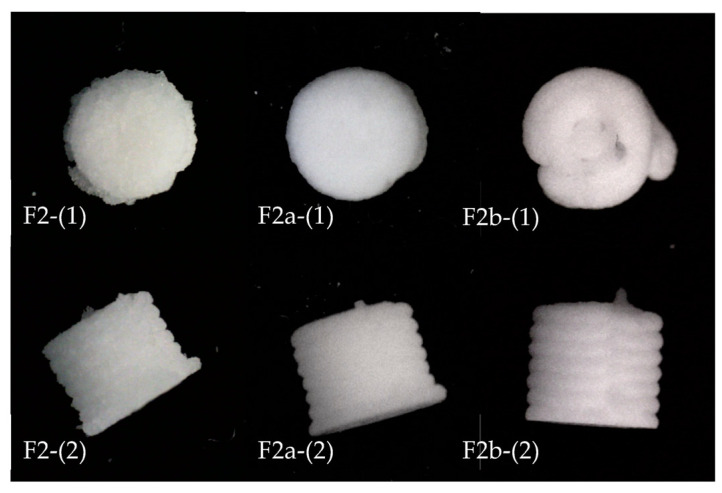
Appearance of F2a and F2b compared with F2 ODMTs (post-drying for 24 h); F2-(1), F2a-(1), F2b-(1): top view; F2-(2), F2a-(2), F2b-(2): side view.

**Figure 7 pharmaceutics-15-00250-f007:**
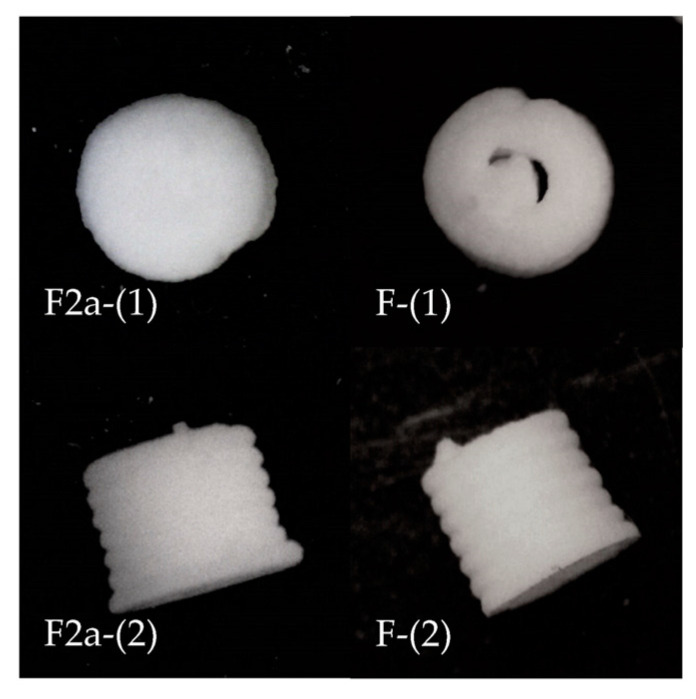
Appearance of F2a compared with F ODMTs (post-drying for 24 h); F2a-(1), F-(1): top view; F2a-(2), F-(2): side view.

**Figure 8 pharmaceutics-15-00250-f008:**
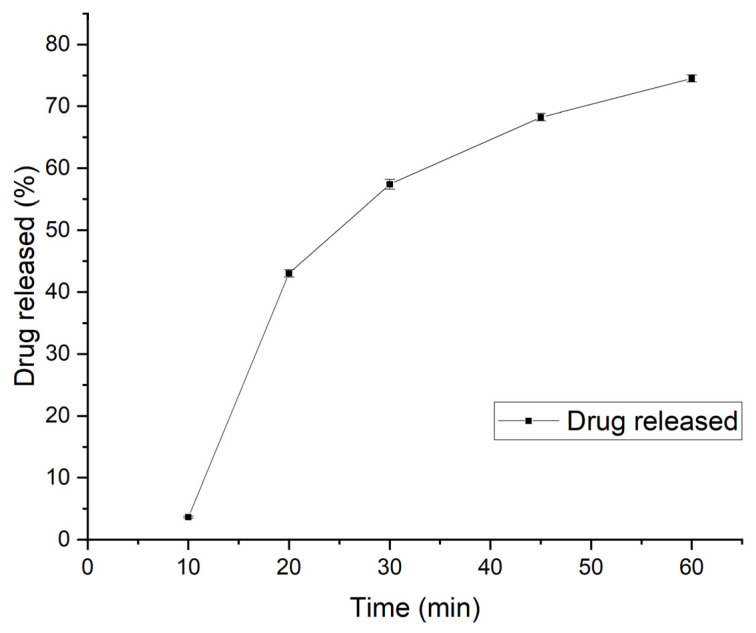
In vitro dissolution profile of CBZ ODMT.

**Figure 9 pharmaceutics-15-00250-f009:**
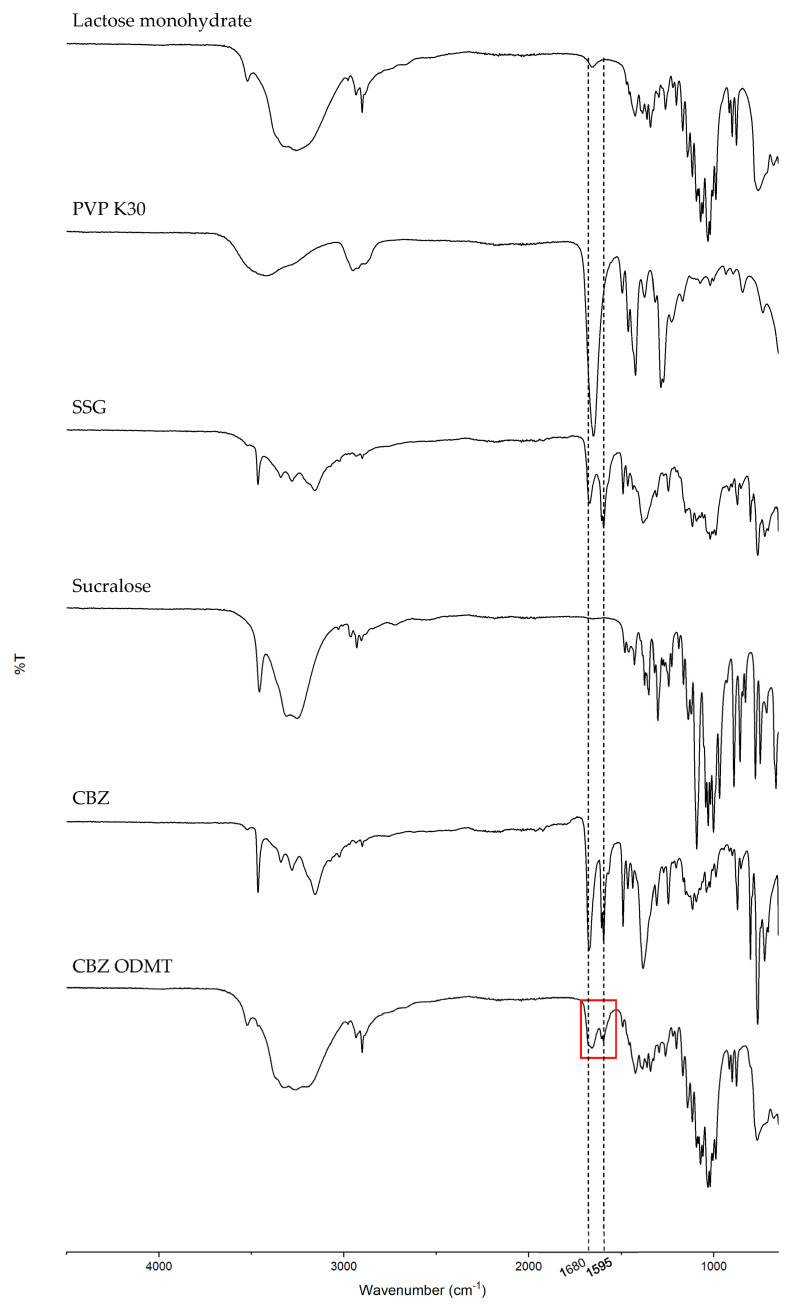
FTIR spectra of lactose monohydrate, PVP K30, SSG, sucralose and drug-loaded ODMTs; the red frame highlighted overlapping peaks of C-O group stretching and C=C group stretching.

**Figure 10 pharmaceutics-15-00250-f010:**
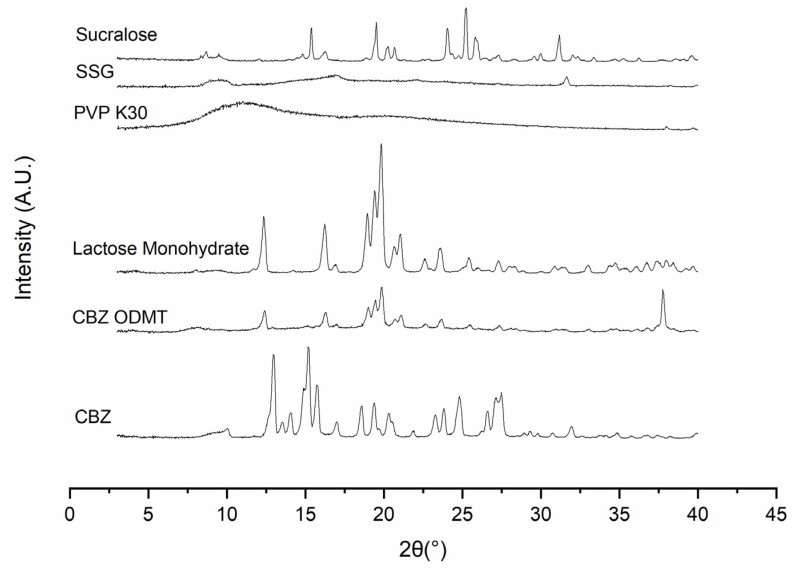
XRD diffraction patterns of lactose monohydrate, PVP K30, SSG, sucralose and drug-loaded ODMTs.

**Figure 11 pharmaceutics-15-00250-f011:**
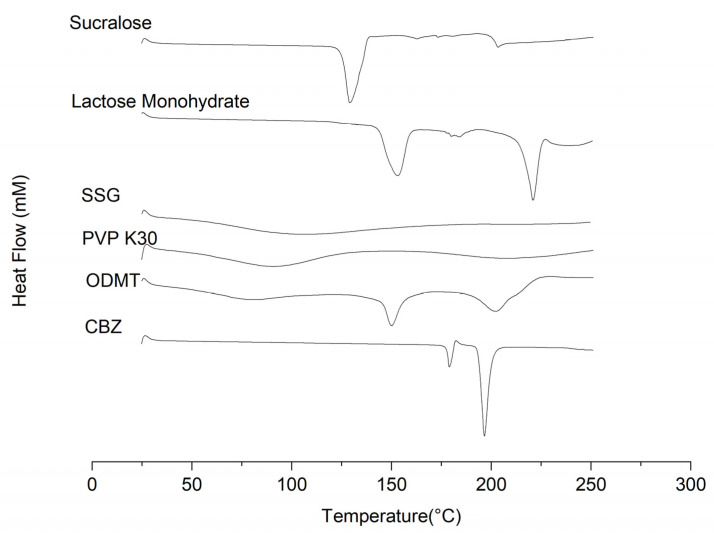
DSC thermogram of lactose monohydrate, PVP K30, SSG, sucralose, drug-loaded wet mass and drug-loaded ODMTs.

**Figure 12 pharmaceutics-15-00250-f012:**
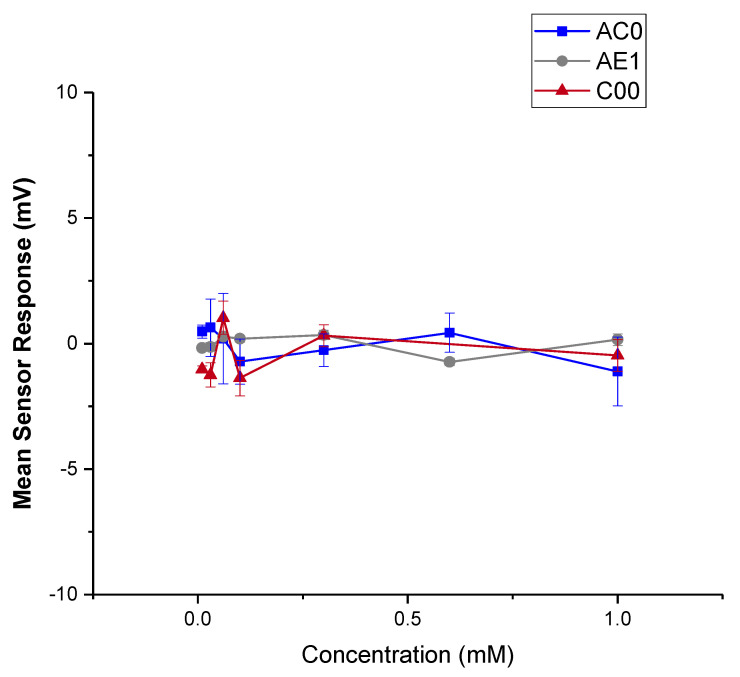
Sensor response curve of pure CBZ in 10 mM KCl (with 10% *v*/*v* ethanol); Mean responses of the AC0, AE1 and C00 sensors reflecting initial taste.

**Figure 13 pharmaceutics-15-00250-f013:**
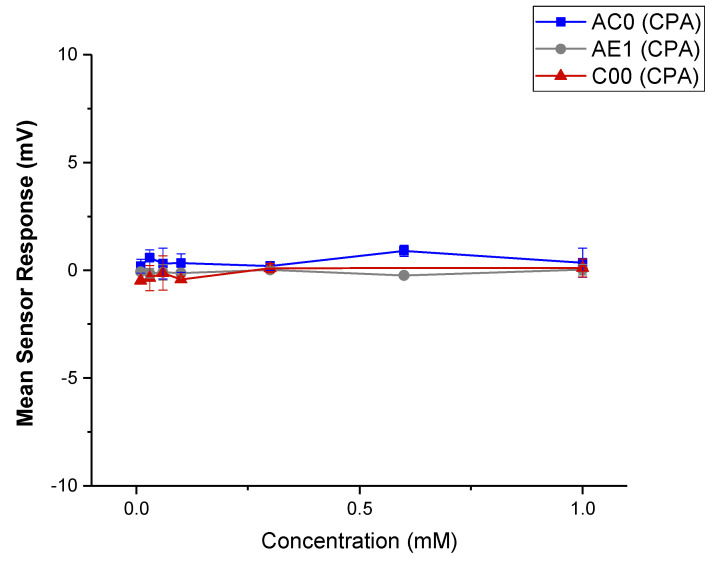
Sensor response curve of pure CBZ in 10 mM KCl (with 10% *v*/*v* ethanol); mean responses of the AC0, AE1 and C00 sensors reflecting aftertaste (CPA).

**Figure 14 pharmaceutics-15-00250-f014:**
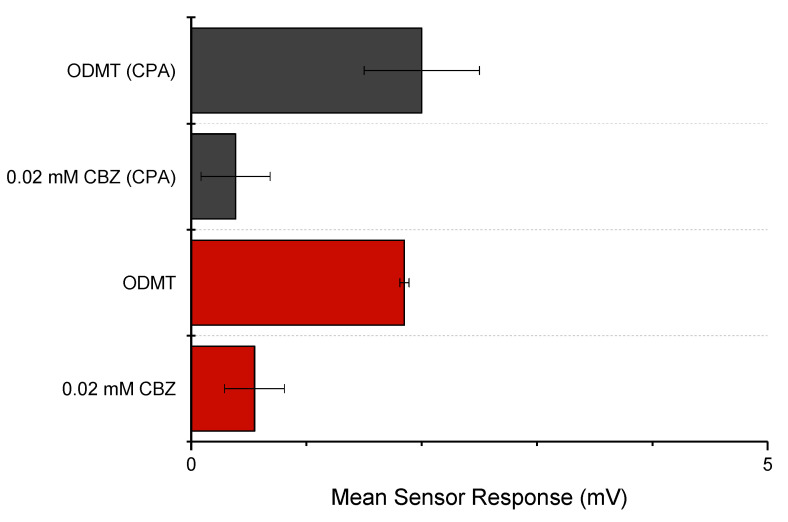
AC0 response of CBZ ODMT reacting compared with pure CBZ, including initial taste and CPA.

**Table 1 pharmaceutics-15-00250-t001:** An overview of the response surface design for optimising CBZ ODMT formulations.

Variables	Specification
Input factor *X1*: Superdisintegrant concentration (*w*/*w*)	Lowest level: 30%; Medium level: 40%; Highest level: 50%
Input factor *X2*: Binder concentration (*w*/*w*)	Lowest level: 2.5%; Medium level:5%; Highest level: 7.5%
Output factor *Y1*: Disintegrating time (s)	Maximum: 300 s; No minimum value; Optimal: minimise
Output factor *Y2*: Hardness	Optimal: maximise

**Table 2 pharmaceutics-15-00250-t002:** Matrix of CCD for CBZ ODMT formulations.

Formulation Code	Superdisintegrant Concentration (*w*/*w*)	Binder Concentration (*w*/*w*)
F1	30%	2.5%
F2	40%	5%
F3	35%	2.5%
F4	40%	2.5%
F5	30%	5%
F6	35%	5%
F7	30%	7.5%
F8	35%	5%
F9	35%	7.5%
F10	40%	7.5%

**Table 3 pharmaceutics-15-00250-t003:** Formula of each batch of CBZ ODMTs.

Formulation Code	Ac-Di-Sol (*w*/*w*)	Kollidon 30 (*w*/*w*)	Lactose Monohydrate (*w*/*w*)	Sucralose (*w*/*w*)	Water (mL)
F1	30%	2.5%	62.5%	5%	5
F2	40%	5%	50%	5%	5
F3	35%	2.5%	57.5%	5%	5
F4	40%	2.5%	52.5%	5%	5
F5	30%	5%	60%	5%	5
F6	35%	5%	55%	5%	5
F7	30%	7.5%	57.5%	5%	5
F8	35%	5%	55%	5%	5
F9	35%	7.5%	52.5%	5%	5
F10	40%	7.5%	47.5%	5%	5

**Table 4 pharmaceutics-15-00250-t004:** Critical parameters and their value in SSE fabrication.

Parameter	Value
Paste container	3 mL syringe
Printing substrate	Petri dish
Nozzle diameter	0.41 mm
Temperature	25 °C
Pressure	Minimum value for printing, no more than 200 kPa
Speed	5 mm/s
Infill pattern	Concentric
Infill density	50%

**Table 5 pharmaceutics-15-00250-t005:** Different lipid membrane sensors used in the E-tongue testing.

Sensor	Information
C00	Positively charged membrane, testing acidic bitterness
AE1	Positively charged membrane, testing astringency
AC0	Negatively charged membrane, testing basic bitterness

**Table 6 pharmaceutics-15-00250-t006:** Minimum pressure required for SSE printing of each DoE formulation.

Formulation Code	Minimum Pressure Required for SSE Printing (kPa)
F1	100
F2	190
F3	170
F4	>200 ^1^
F5	100
F6	>200
F7	200
F8	>200
F9	200
F10	>200

^1^ F4 (and same with F6, F8 and F10) was not printable even with the extrusion pressure reaching its maximum value of 200 kPa, hence its minimum pressure required is marked as >200.

**Table 7 pharmaceutics-15-00250-t007:** Characterisation of ODMT formulation in DoE.

Formulation Code	Diameter (mm)	Height (mm)	Weight (mg)	Fidelity Factor	Hardness (N)	Disintegrating Time (s)
F1	3.003 ± 0.210 ^1^	2.258 ± 0.163	7.96 ± 1.65	0.760 ± 0.132	49.6 ± 12.31	228 ± 22.77
F2	3.027 ± 0.181	2.239 ± 0.124	9.62 ± 1.66	0.765 ± 0.111	61.3 ± 14.35	175 ± 8.11
F3	3.320 ± 0.262	2.518 ± 0.195	15.30 ± 2.15	1.034 ± 0.183	101.2 ± 13.20	328 ± 18.54
F5	3.342 ± 0.117	2.362 ± 0.121	15.38 ± 1.25	0.977 ± 0.064	81.9 ± 10.65	329 ± 12.54
F7	3.514 ± 0.231	2.664 ± 0.119	17.58 ± 2.02	1.222 ± 0.154	98.7 ± 5.18	380 ± 18.87
F9	2.792 ± 0.152	2.226 ± 0.072	8.56 ± 1.33	0.645 ± 0.078	57.1 ± 8.14	215 ± 23.85

^1^ All data are presented as mean ± SD; n (sample size) of diameter, height, weight = 25, n of hardness = 10, n of disintegrating time = 6.

**Table 8 pharmaceutics-15-00250-t008:** Summary of fit of the standard least squares regression model used to fit two responses separately in DoE analysis.

Response	R Square	Mean of Response	Observations
Disintegrating time	1	275.8333	6
Hardness	1	74.96667	6

**Table 9 pharmaceutics-15-00250-t009:** Differences in preparation and fabrication via SSE of F2a, F2b and F2.

Formulation Code	Superdisintegrant	Solid:Liquid Ratio in Paste	Extrusion Pressure
F2	Ac-Di-Sol (40% *w*/*w*)	2 mg solid:5 mL water	190 kPa
F2a	SSG (40% *w*/*w*)	2 mg solid:2.6 mL water	170 kPa
F2b	D-Mannitol (30% *w*/*w*) + SSG (10% *w*/*w*)	4 mg solid:2.1 mL water	150 kPa

**Table 10 pharmaceutics-15-00250-t010:** Characterisation results of F2a and F2b compared with F2.

Formulation Code	Diameter (mm)	Height (mm)	Fidelity Factor	Weight (mg)	Hardness (N)	Disintegrating Time (s)
F2	3.027 ± 0.181 ^1^	2.239 ± 0.124	0.765 ± 0.111	9.62 ± 1.66	61.3 ± 14.35	175 ± 8.11
F2a	3.341 ± 0.107	2.804 ± 0.064	1.161 ± 0.075	19.63 ± 1.39	53.3 ± 6.65	80 ± 8.22
F2b	3.136 ± 0.321	2.870 ± 0.089	1.056 ± 0.169	21.23 ± 1.25	39.2 ± 8.00	86 ± 9.38

^1^ All data are presented as mean ± SD; n (sample size) of diameter, height, weight = 25, n of hardness = 10, n of disintegrating time = 6.

**Table 11 pharmaceutics-15-00250-t011:** Differences in preparation and fabrication via SSE of F and F2a.

Formulation Code	Composition	Solid:Liquid Ratio in Paste	Extrusion Pressure
F2a	-	2 mg solid:2.6 mL water	170 kPa
F	F2a + 100 mg CBZ	2.1 mg solid:2.6 mL water	110 kPa

**Table 12 pharmaceutics-15-00250-t012:** Characterisation results of F compared with F2a.

Formulation Code	Diameter (mm)	Height (mm)	Fidelity Factor	Weight (mg)	Hardness (N)	Disintegrating Time (s)
F2a	3.341 ± 0.107 ^1^	2.804 ± 0.064	1.161 ± 0.075	19.63 ± 1.39	53.3 ± 6.65	80 ± 8.22
F	2.490 ± 0.081	2.465 ± 0.060	0.567 ± 0.043	10.71 ± 0.50	18.5 ± 3.61	84 ± 3.40

^1^ All data are presented as mean ± SD; n (sample size) of diameter, height, weight = 25, n of hardness = 10, n of disintegrating time = 6.

**Table 13 pharmaceutics-15-00250-t013:** Content assay results of CBZ ODMT formulation.

Sample	Mass of ODMT Powder (mg)	Amount of Drug (mg)	Composition of Drug in the Powder (% *w*/*w*)	Drug-Loading (%)
1	10.7	0.4594	4.29	90.16
2	11.2	0.4835	4.31	90.66
3	10.5	0.4543	4.33	90.86
Mean ± SD		0.4657 ± 0.01	4.31 ± 0.02	90.56 ± 0.29

## Data Availability

All data available in this paper.

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
