# Peer review of "Formulation and Characterisation of Carbamazepine Orodispersible 3D-Printed Mini-Tablets for Paediatric Use"

_pharmaceutics, 2023, doi:10.3390/pharmaceutics15010250_

Round 1

Reviewer 1 Report

Dear Authors,

It was a real pleasure to review your manuscript. It describes a very interesting research, well designed and well written. I have some remarks and suggestions to consider and correct before publication, however, most of them requires only slight modifications to the text:

1.       Page 2, line 64 – the term “orodispersible tablets” is used only in European Pharmacopoeia. The USP rather calls these forms as orally disintegrating tablets.

2.       Page 4, table 1 and table 2 – concentration of superdisintegrant is different in these two tables. In table 1, it is 30%, 40% and 50%, while in table 2, it is 30%, 35% and 40%.

3.       Page 5, first paragraph – There is no information on the weight of tablet mass, to which 100 mg of carbamazepine was added. I could guess from the result section, that it was 2 g, but it should be described in the Methods section.

4.       Page 6, line 207-215 – you have stated that disintegration test was made according to the USP requirements. However, you have used the wire cloth with 1 mm mesh size. I suggest to state that it was different than the mesh size required in the USP (which is 2 mm).

5.       Page 6, line 218 – the size of a single batch was 25 tablets (page 5, line 180). How can 48 tablets from each batch used for the friability test?

6.       Page 6, line 238 – it is unclear whether 3 minitablets were placed in one basket, or individually put to three different baskets.

I suggest also to add the information here, that the dissolution test was performed in static conditions, i.e. without rotation of the basket.

7.       Page 8, line 326 – the batch size is expressed as 50 tablets, here, which is not consistent with an information from page 5, line 180 (25 tablets). Please clarify this.

It would be beneficial to add information on the printing process yield, here. How much of the extruded mass was wasted during the preparation process, and what was the weight of prepared mass?

8.       Page 9, line 343 – the final bracket symbol is missing, after ‘200 kPa’.

9.       Page 9, line 348-350 – it seems that the sentence doesn’t contain a verb. Shouldn’t it be like “However, during the actual preparation, the viscoelastic properties varied due to the different composition of excipients.”?

10.   Page 9, figure 2 – it is unclear which tablets are presented on the figure. I suggest to name them in the same manner like in the DoE, i.e. F1, F2, F3, F5, F7, F9, instead of 1, 2, 3, etc.

For example, as far as I understood, the picture 4a and 4b represents formulation F5, while 6a and 6b are for formulation F9. It is a little bit confusing for the reader.

11.   Page 10, lines 389 – 391 – according to the European Pharmacopoeia requirements, the mesh size for the disintegration test is 2 mm. There is no other value reported in the work of Eduardo, Ana and Jose (reference no. 20). Where does the value 3 mm comes from? Isn’t it a mistake?

12.   Page 12, table 9 – You have reported that the solid to liquid ratio in the paste was 2 mg or 4 mg to 2.1-5 mL of water. It seems like there should be 2 g and 4 g. The same is in the table 11.

13.   Page 13, table 10 and figure 6 – formulation code for modified formulation is F2a, and F2b, so on the figure 6, there are symbols like F2-a (which is a picture of formulation F2) and F2a-a (which is a picture of formulation F2a). It may be very confusing. I suggest changing the coding system for the formulations, at least for F2A and F2B.

14.   Page 13, line 460 – should be “Ac-Di-Sol” instead “Ac-D-Sol”.

15.   Page 14, table 11 – no information on the amount of mass F2a, which was mixed with 100 mg of carbamazepine. I guess that it was 2 g, but it is not clear from the table or the text, especially that the solid to liquid ratio in that table seems to be erroneously expressed in milligrams instead of grams.

16.   Page 16, lines 528-534 – you have pointed out that the weakness of the dissolution analysis was no media stirring during the study. Probably, it was the major cause of extended CBZ release. Taking into account the solubility of CBZ, which is 0.152 mg/mL (15.2 mg in the dissolution vessel that you have used – 100 mL), despite the low solubility of this compound, you have met sink conditions (the amount of CBZ in one minitablet was 0.46 mg).

Therefore, addition of the SLS to dissolution medium, in this case is not justified, since the basic role of the addition of surfactant is to achieve the sink conditions in the case of poorly soluble drugs.

17.   Page 17, figure 9 – images have symbols A-F, which might be mistakenly taken as a formulation symbols. I suggest adding an information that it is formulation F. The images can be then indicated as a, b, c, d, etc.

Reviewer 2 Report

The authors attempted to prepare OD minitablet containing carbamazepine by using semi-solid 3D printer. I think this manuscript is important for the pediatric patients, and it includes good information about DoE.

Overall, disintegration of tablet is not so fast (80s) compared to conventional OD tablets (30s). This is a comment.

OD tablet needs hardness to some extent, how did you set the criteria about the hardness?

Fig. 8. This dissolution profile is weird. There is time lag, so explanation about the result is necessary. Then, the drug did not release completely. The author should take longer time point. Why did you choose this drug formulation?   

Regarding the evaluation of taste by taste sensor, the author could not get effective response (e.g. Fig. 13, 14). I know that taste sensor is not almighty, although it is used for the assessment of bitterness. The reviewer think that these descriptions will inhibit the whole discussion. I recommend the moving the data into supplementary section.

Fig. 15. The maximum of axis is 10 mV, in contrast, the control exhibited 2-3 mV. It is weird for me.

3D printing of drug formulation using Cellink 3D bio printer is published as follow. These references are probabaly useful for you. Can you cite them with small explanation?

- Tagami et al., Fabrication of Naftopidil-Loaded Tablets Using a Semisolid Extrusion-Type 3D Printer and the Characteristics of the Printed Hydrogel and Resulting Tablets. JPS. 2019.

- Panraksa et al., Tablet-in-Syringe’: A Novel Dosing Mechanism for Dysphagic Patients Containing Fast-Disintegrating Tablets Fabricated Using Semisolid Extrusion 3D Printing. Pharmaceutics. 2022.

- Chatzitaki et al., Semi-solid extrusion 3D printing of starch-based soft dosage forms for the treatment of paediatric latent tuberculosis infection. JPP. 2022

Reviewer 3 Report

The research require a major revision before consideration for publication following the below mention comments.

1. The authors stated the highest pressure of instrument is 200 kPa. Formulations F4, F6, F7, F8, F9, and F10. Authors stated that F4, F6, F8, and F10 reached the upper pressure limit, what about F7 and F9, which also reached the higher limit, how it was formulated and how significantly the results are reproducible for those formulations.

2. Authors suggested to calculate the shape fidelity factors, to understand difference between theoretical value obtained from Tinkercad software and practical values of height and other parameters.

3. Several directly compressed ODT reported with disintegrating time less than 180 s. authors suggested to explain how the current research envisaged is effective compared with directly compressible formulations.

4. Why Ac-Di-Sol was replaced by sodium starch glycolate or D-mannitol after optimising the formulation in statistical design? why not during optimisation these polymers were tested.

5. Suggested to add shift in wavenumber in FTIR

6. A significant shift in CBZ peak observed in DSC, suggested to provide suitable explanation 

7. Suggested to summarise the conclusion not to write discussion again.

8. Authors are suggested to explain in discussion briefly about use of super-disintegrant and why variation in disintegration is high compared to use of same excitants in direct compression technology.

Round 2

Reviewer 3 Report

The Authors have reflected all the suggestions, I recommended that manuscript can be accepted in present form